# Long, Noncoding RNA SRA Induces Apoptosis of β-Cells by Promoting the IRAK1/LDHA/Lactate Pathway

**DOI:** 10.3390/ijms22041720

**Published:** 2021-02-09

**Authors:** Yu-Nan Huang, Shang-Lun Chiang, Yu-Jung Lin, Su-Ching Liu, Yen-Hsien Li, Yu-Chen Liao, Maw-Rong Lee, Pen-Hua Su, Fuu-Jen Tsai, Hui-Chih Hung, Chung-Hsing Wang

**Affiliations:** 1Department of Life Sciences, National Chung Hsing University, Taichung 402, Taiwan; yunanhuang83@gmail.com (Y.-N.H.); hchung@dragon.nchu.edu.tw (H.-C.H.); 2Division of Genetics and Metabolism, Children’s Hospital of China Medical University, Taichung 402, Taiwan; 3Department of Medical Laboratory Science, I-Shou University, Kaohsiung 824, Taiwan; chimpanzee99999@gmail.com; 4Cardiovascular and Mitochondrial Related Disease Research Center, Hualien Tzu Chi Hospital, Buddhist Tzu Chi Medical Foundation, Hualien 970, Taiwan; u9882851@gmail.com; 5Department of Medical Research, Children’s Hospital of China Medical University, Taichung 404, Taiwan; t0595@mail.cmuh.org.tw; 6Department of Chemistry, National Chung Hsing University, Taichung 420, Taiwan; liwenshin@dragon.nchu.edu.tw (Y.-H.L.); mrlee@dragon.nchu.edu.tw (M.-R.L.); 7Instrument Center, Office of Research and Development, National Chung Hsing University, Taichung 420, Taiwan; ycliao@dragon.nchu.edu.tw; 8Department of Pediatrics, Chung Shan Medical University Hospital, Taichung 412, Taiwan; ninaphsu@gmail.com; 9School of Medicine, Chung Shan Medical University, Taichung City 40201, Taiwan; 10Genetic Center, Department of Medical Research, China Medical University Hospital, Taichung 40447, Taiwan; d0704@mail.cmuh.org.tw; 11School of Medicine, China Medical University, Taichung 404, Taiwan

**Keywords:** LncRNA SRA, LDHA, lactate, β-cell, type 1 diabetes mellitus

## Abstract

Long non-coding RNA steroid receptor RNA activators (LncRNA SRAs) are implicated in the β-cell destruction of Type 1 diabetes mellitus (T1D), but functional association remains poorly understood. Here, we aimed to verify the role of LncRNA SRA regulation in β-cells. LncRNA SRAs were highly expressed in plasma samples and peripheral blood mononuclear cells (PBMCs) from T1D patients. LncRNA SRA was strongly upregulated by high-glucose treatment. LncRNA SRA acts as a microRNA (miR)-146b sponge through direct sequence–structure interactions. Silencing of lncRNA SRA increased the functional genes of Tregs, resulting in metabolic reprogramming, such as decreased lactate levels, repressed lactate dehydrogenase A (LDHA)/phosphorylated LDHA (pLDHA at Tyr10) expression, decreased reactive oxygen species (ROS) production, increased ATP production, and finally, decreased β-cell apoptosis in vitro. There was a positive association between lactate level and hemoglobin A1c (HbA1c) level in the plasma from patients with T1D. Recombinant human interleukin (IL)-2 treatment repressed lncRNA SRA expression and activity in β-cells. Higher levels of lncRNA-SRA/lactate in the plasma are associated with poor regulation in T1D patients. LncRNA SRA contributed to T1D pathogenesis through the inhibition of miR-146b in β-cells, with activating signaling transduction of interleukin-1 receptor-associated kinase 1 (IRAK1)/LDHA/pLDHA. Taken together, LncRNA SRA plays a critical role in the function of β-cells.

## 1. Introduction

Type 1 diabetes mellitus (T1D) develops as a result of various genetic and environmental factors, and is clinically characterized as the progressive loss of β-cell mass and absolute insulin deficiency over time [1]. In 2017, the global estimate for T1D in children and adolescents (<20 years) was approximately 1.1 million, with 132,600 new cases per year [2]. Strategies for early therapy and prevention of T1D are still emerging. Metabolic reprogramming has become an important sign of T1D development. T1D involves unique metabolic pathway changes, such as decreased aerobic glycolysis, diminished glucose utilization, reduced efficiency in ATP generation, upregulated fatty acid oxidation, and augmented oxidative phosphorylation [3,4]. T1D is characterized by hyperglycemia-induced cellular damage and decreases in regulatory T cells (Tregs) that further interfere with the function of β-cells [5]. Abnormal metabolism is the leading cause of complications in patients with poorly regulated T1D.

Long, non-coding RNA (lncRNA) regulates multiple physiological functions that govern epigenetic modifications, transcriptional regulation, posttranscriptional regulation, and small ncRNA processing during cellular development and homeostasis [6]. In the last 10 years, several in vitro and in vivo studies have demonstrated the possible role of lncRNA in the pathogenesis of diabetes, including processes like pancreatic β-cell function and insulin secretion (Insulin-Like Growth Factor 2 Antisense (IGF2-AS) [7,8], βlinc1 [9], and Maternally Expressed 3 (MEG3) [10,11]), as well as glucose homeostasis and insulin sensitivity (H19 [12] and steroid receptor RNA activator (SRA) 1 [13]). lncRNA SRAs were first investigated as transcriptional coactivators [14], and have been shown to be involved in glucose homeostasis and protein kinase B (AKT)-dependent insulin sensitivity in adipocytes [14,15]. In this study, we investigated candidate lncRNA SRAs in relation to T1D.

miRNAs are small, endogenous, noncoding RNAs that repress mRNA translation by binding to the 3’-untranslated regions (3′-UTRs) of different target genes. As a novel class of posttranscriptional regulatory elements, miRNAs are optimal candidates for the modulation of the immune response and the pancreatic–endocrine system. miRNA (miR)-34a and miR-146 are involved in β-cell apoptosis in pancreatic islets [16]. Notably, miR-146a/b was shown to maintain differentiated T cell lineages in murine immune regulation [17]. MiR-146a may be an important modulator of cell differentiation and innate immunity, as well as some autoimmune diseases [18,19]. lncRNA acts as an miRNA sponge through direct sequence–structure interactions [12,20]. However, no studies have investigated the interaction between highly conserved miR-146b and lncRNA SRAs in immune regulation relative to T1D.

In this paper, we show that the plasma expression level of lncRNA SRAs increases in patients with T1D. Subsequently, the level of miR-146b, an miRNA that is highly conserved in primates, decreased in high-glucose conditions, while the level of lncRNA SRAs increased. In addition, lncRNA SRA is negatively regulated by miR-146b. Knockdown of SRA impaired lactate dehydrogenase A (LDHA) and the phosphorylation of LDHA (Tyr10), decreased reactive oxygen species (ROS) production, repressed lactate secretion, and rescued β-cell apoptosis. Finally, interleukin (IL)-2 treatment might antagonize these processes to induce the development of T1D. Together, these data showed that the lncRNA SRA and miR-146b are reciprocally regulated in T1D.

## 2. Results and Discussion

### 2.1. Demographic Characteristics of T1D Patients

The demographic characteristics of the young patients with T1D are shown in Table 1. The 25 participants with T1D had a mean age of 18.4 ± 5.2 years (40% male). Participants had a mean hemoglobin A1c (HbA1c) of 9.0% ± 1.8% and a C-peptide level of 0.4 ± 0.2 ng/mL. There were no differences in hemoglobin, hematocrit, complete blood count, total cholesterol, high-density lipoprotein, or low-density lipoprotein between males and females.

### 2.2. The Effect of High Glucose on Candidate lncRNAs/miRNAs in CD4+ MOLT4 Tregs and the Expression of lncRNA-SRA/miR-146b in the Plasma of Patients with T1D

To determine whether lncRNAs are involved in CD4+ Treg-mediated reprogramming in T1D, we determined glucose-induced lncRNA expression in CD4+ Tregs using a qPCR-based, disease-related lncRNA array (Systems Biosciences). Deregulated lncRNAs were determined in CD4+ Tregs after 72 h of high-glucose conditions (high-glucose conditions, 2.5-fold change glucose compared to normal conditions; normalized to normal conditions). The results revealed high glucose-induced differential expression of lncRNAs: 10 lncRNAs were upregulated (>20-fold). Among the upregulated lncRNAs, MEG3 (~95), prostate cancer associated transcript 32 (PCAT-32) (~39), LincRNA-very low density lipoprotein receptor (VLDLR) (~36), DiGeorge syndrome critical region gene 5 (DGCR5) (~30), ST7 overlapping transcript 3 (ST7OT3) (~26), TU 0017629 (~25), BC043430 (~25), AK023948 (~25), PSF-inhibiting RNA (~21), and SRA (~20) were identified (Figure 1A). We ranked the lncRNAs from the disease-related lncRNA array with the top 20-fold changes, and identified 10 lncRNAs related to glucose metabolism based on the literature to select the candidate lncRNAs for further study (Figure 1B and Appendix A). We measured the expression levels of three candidate lncRNAs (IGF2AS, MEG3, and SRA) under high-glucose and normal conditions. As shown in Figure 1C, compared to those in the normal (N) conditions, the expression levels of the lncRNAs MEG3 and SRA were significantly increased in the high-glucose conditions (*p* < 0.001), and lncRNA SRA was strongly expressed. To investigate possible interactions between lncRNA SRA and miRNAs, we identified the top 10 candidate miRNAs by bioinformatic predictions (Appendix A). Four highly-conserved miRNAs (miR-146b, miR-148a, miR-203, and miR-103a) were tested in the in vitro analysis (Figure 1D). Compared to the normal conditions, in the high-glucose conditions, the expression of three miRNAs (miR-146b, miR-148a, and miR-203) was downregulated, and miR-103a expression was upregulated (*p* < 0.001; Figure 1E). Furthermore, we enrolled 25 young patients who were newly diagnosed with T1D in childhood (mean age at disease onset was 9.5 years) (Table 1). The fasting glucose and HbA1c levels in the enrolled population were improved with clinical therapies compared to those at the age of onset, but were still out of the normal range. We further assessed the expression levels of plasma miR-146b and lncRNA SRA in T1D patients and healthy controls, and increased SRA levels (Figure 1F) and decreased miR-146b levels (Figure 1G) were observed. These data suggest that lncRNA SRA and miR-146b were reciprocally expressed in T1D patients.

### 2.3. LncRNA SRA Is a Direct Target of miR-146b and Regulates IRAK1-AKT-S6K1 Signaling

To understand the effect of SRA and miR-146b on the interleukin-1 receptor-associated kinase 1 (IRAK1)/protein kinase B (AKT)/S6K1 signaling, we performed a reporter assay, transfected with shSRA and miR-146b mimics to the stable CD4+ Tregs cells, and immunoblot analysis in the study. The designed lncRNA SRA sequence (transcript variant 4) had two putative target sites for miR-146b at seed positions +219 and +1122 with a 6- to 9-mer match (Figure 2A). Two wild-type (WT) sequences were cloned downstream of the firefly luciferase coding region as short and long lncRNA SRA sequences in parallel with mutant derivatives, which lacked the miRNA response elements (Figure 2B). We further assessed the luciferase activity of the short and long lncRNA SRA sequences that were transfected into Jurkat T cells in the presence of the scrambled-miR, miR-146b mimic, or miR-146b antagomir (Figure 2C). The miR-146b mimic decreased the level of lncRNA SRA through both putative binding sites, and it showed a greater inhibitory effect at seed position +219 than at seed position +1122. Moreover, the effect was reversed by the miR-146b antagomir. MiR-146b affected the expression of lncRNA SRA, and lncRNA SRA inhibited miR-146b expression in a dose-dependent manner (Figure 2D). These results show that SRA controls miR-146b, and that miR-146b also administers SRA expression; therefore, the repressive activities of both are reciprocally regulated. To investigate the possible interaction between lncRNA SRA and miR-146b, we performed a crossover study. First, scrambled and miR-146b mimics were transfected into T cells, and the expression level of lncRNA SRA was assessed. The endogenous SRA expression level was significantly lower at 24 h post-transfection in the cells transfected with the miR-146b mimic than in the cells transfected with the scrambled mimic (Figure 2E). The effects of the miR-146b mimic and shSRA knockdown on the IRAK1/AKT/mammalian Target Of Rapamicyn (mTOR) signaling network were further investigated using select mediators in the immunoblotting analysis. The results showed that the miR-146b mimic decreased upstream IRAK1 signaling and the downstream phosphorylation of the AKT and S6 ribosomal proteins (Figure 2F). In contrast, the endogenous miR-146b level was significantly increased at 24 h post-transfection in the cells transfected with the SRA knockdown shRNA vector, compared to the cells transfected with the shRNA control vector (Figure 2G). Moreover, the increase in endogenous miR-146b expression induced by the transfection of the shSRA vector was similar to that induced by the transfection of the miR-146b mimic (Figure 2H). These results show that miR-146b controls IRAK1/AKT/mTOR signaling, and that this repressing activity is modulated by lncRNA-SRA, indicating that IRAK1/AKT/mTOR signaling also participates in crosstalk with lncRNA-SRA through competition for miR-146b.

### 2.4. LncRNA SRA Suppresses CD4+ Treg Function

To identify the impact of lncRNA SRA on Treg function, we assessed the following Treg signature genes: Foxp3, TNFRSF18, IL2RA, IKZF2, and IKZF4. CD4+ MOLT4 Treg stable cell lines (short hairpin Control, shCon, shSRA-1, and shSRA-2) were established and then analyzed with RT-qPCR (Figure 3A) for corresponding changes in lactate secretion after SRA inhibition (Figure 3B). To confirm the results of the many Treg signature genes after SRA manipulation, we next profiled the ability of SRA knockdown to reciprocally turn on the mRNA levels of the same genes. Using RT-qPCR, we confirmed that SRA knockdown at the gene level caused increased mRNA expression of Treg signature genes (FOXP3, TNFRSF18, and IL2RA) (Figure 3C). The results showed that SRA represses Treg function by downregulating the mRNA expression levels of Treg signature genes.

### 2.5. LncRNA SRA Induces MIN6 β-Cell Metabolic Reprogramming and the IRAK1/LDHA/pLDHA Signaling Pathway

Further confirmation of the hypothesis that SRA inhibition switches cellular metabolism towards aerobic glycolysis came from transcription profiling studies (RT-qPCR) performed on MIN6 β-cells after SRA inhibition (shCon, shSRA-1, and shSRA-2; Figure 4A). Figure 4B shows the sequential steps of the metabolic pathway, from transport across the cell membrane, glycolysis, the pentose phosphate pathway, lactate production, glutaminolysis, fatty acid metabolism, and finally, entry into the Krebs cycle, for both enzymes and metabolites. We examined alterations in the mRNA levels of key metabolic enzymes in MIN6 β-cells (Figure 4B) and the corresponding changes in lactate secretion after SRA inhibition (Figure 4C,D). Cells were collected from different MIN6 β-cell lines (shCon, shSRA-1, and shSRA-2) and analyzed by RT-qPCR. SRA inhibition upregulated the expression of early glucose transporters and enzymes involved in early glycolysis (Hexokinase 2, HK2 and Phosphofructokinase liver type, PFKL) (Figure 4B). SRA inhibition downregulated enzymes involved in late glycolysis (LDHA) (Figure 4B). To further confirm this shift to accelerated glucose utilization through aerobic glycolysis, we studied the lactate production rates after SRA inhibition in cultured MIN6 β-cell lines (shCon, shSRA-1, and shSRA-2). To further characterize metabolic changes in lactate production induced by SRA silencing, we used staining of IRAK1, an inducer of the inflammatory response that impairs insulin signaling, and LDHA/pLDHA, a key enzyme of lactate production, to monitor the protein expression of IRAK1 and LDHA/pLDHA in β-cells (shCon vs shSRA) (Figure 4E). Mechanistically, lactate production occurs through phosphorylation of LDHA (Tyr10) and subsequent conversion of pyruvate to lactate, which mediates metabolic homeostasis. Consistent with this idea, SRA suppression downregulated both LDHA and phosphorylated LDHA (Tyr10) (Figure 4E). These data support an SRA-driven process of a shift in lactate metabolism.

### 2.6. LncRNA SRA Silencing Induces ATP Production, Represses ROS Levels, and Inhibits MIN6 β-Cell Apoptosis

The above findings led us to examine the influence of SRA silencing on metabolic reprogramming and its influence on lactate production in β-cell models. To gauge the impact of SRA inhibition on the effect of ATP production, we measured the cellular ATP content. The ratio of ATP increased (Figure 5A), indicating that SRA inhibition rescues oxidative phosphorylation and contributes to efficient ATP generation in β-cells. In addition, we measured the ROS to gauge the impact of SRA inhibition on cellular antioxidant capacity. We indeed observed that in response to SRA silencing, ROS progressively decreased (Figure 5B). Using spheroids obtained from β-cell cultures, we found that SRA silencing reduced the apoptosis of β-cells (Figure 5C). Propidium iodide (PI) and annexin V staining confirmed that SRA promoted cell death to a larger extent than shCon in β-cells, while ATP detection revealed that the latter exerted more significant antiapoptotic effects than shCon on β-cells (Figure 5A,C). Collectively, these data indicate that ROS and apoptosis repression induced by SRA inhibition are sufficient to trigger β-cell survival.

### 2.7. The Effect of rhIL-2 Treatment on the Expression of lncRNA SRA, Metabolic Pathways in MIN6 β-Cells, and Lactate Associated with Poor Clinical Outcome

IL-2 therapy has been shown to promote the expansion of Tregs in T1D and to control autoimmune diseases [21]. Here, we briefly tested the therapeutic efficacy of IL-2 treatment on the regulation of endogenous levels of lncRNA SRA. The results showed that IL-2 treatment significantly decreased SRA in β-cells (Figure 6A). Our data also showed that IL-2 upregulated several key metabolic enzymes identified by RT-qPCR analysis, including glucose transporter 1 (GLUT1), HK2, and PFKL (glucose transport and glycolysis); glucose 6 phosphate dehydrogenase (G6PD), transaldolase 1 (TALDO1), and transketolase (TKT) (pentose phosphate pathway); malate dehydrogenase 2 (MDH2) and succinate dehydrogenase complex flavoprotein subunit a (SDHA) (TCA cycle); glutamic oxaloacetic transaminase 2 (GOT2), glutamic pyruvic transaminase 2 (GPT2), glutamate dehydrogenase 1 (GLUD1), and glutaminase 1 (GLS1) (glutaminolysis); pyruvate dehydrogenase E1 α subunit (PDHA) and pyruvate carboxylase (PC) (pyruvate metabolism); ATP citrate lyase (ACLY) (fatty acid biosynthesis); solute carrier family 16 member 1 (SLC16A1) and SLC16A3 (lactate transport); and carnitine palmitoyltransferase 1A (CPT1A) (fatty acid oxidation) (Figure 6B). Importantly, IL-2 reduced lactate production in β-cells (Figure 6C). Notably, IL-2 treatment in β-cells did not cause changes in LDHA levels (Figure 6B). Thus, it is possible to change the protein expression level. These data indicate that IL-2 treatment might improve the status of SRA in patients with T1D. We next explored lactate production in freshly isolated plasma from patients with poorly regulated T1D. We found that increasing lactate levels were positively associated with a clear increase in HbA1c levels (Figure 6D). Token together, these data characterize potentially therapeutic targets that could restore the function of metabolic-responsive β-cells by augmenting the IL-2 pathway and be maintained in patients with poorly regulated T1D.

## 3. Discussion

In this study, we report that the lncRNA SRA-mediated IRAK1/LDHA/pLDHA/lactate pathway in β-cells is involved in the possible progression of T1D. We found that compared to normal glucose conditions, high glucose conditions increased the level of lncRNA SRA in the cells. Additionally, the expression level of lncRNA SRA was significantly higher in the T1D patients than in the healthy controls. SRA is an intergenic lncRNA that has been shown to act as an RNA coactivator of other type I and type II nuclear receptors, as well as the transcription factor MyoD, in the regulation of muscle differentiation [14]. An SRA knockout transgenic mouse model (SRA^−/−^) displayed improved insulin sensitivity and resistance to developing obesity in high-fat diet conditions [13]. This finding suggests that SRA plays an important role in the development of type 2 diabetes and diabetic complications. Here, according to our findings, we suggest that lncRNA SRA may also be involved in the progression of T1D. We found decreased miR-146b levels in the T1D patients compared with the control subjects. MiR-146 family precursors are found in humans; the mature forms of hsa-miR-146a-5p and hsa-miR-146b-5p have approximately 91% sequence identity, while the sequence identity between hsa-miR-146a-3p and hsa-miR-146b-3p is much lower [22]. Mature miR-146a/b is predicted to base-pair with the 3’-UTRs of the IRAK1 genes (miRDB, http://mirdb.org/ (accessed on 15 January 2021)). IRAK1 mediates the proinflammatory response through interleukin 1 receptor (IL1R)/Toll-like receptor (TLR) signaling, and in intraperitoneal glucose tolerance tests, IRAK1-null mice showed improved muscle insulin sensitivity on a low-fat diet [23]. In this study, we suggest that the potentially protective role of mature miR-146b may contribute to the inhibition of inflammation through IL-1β/IRAK1 signaling in T1D. Serum from recent-onset patients showed differentially expressed genes in the IL-1β signaling pathway through IL1R and TLR signaling, which is associated with the disease process in T1D [24]. The dysregulated, TLR-induced IL-1β signaling in peripheral blood mononuclear cells (PBMCs) was more readily detectable in children with an onset age < 11 years, and this signaling was involved in the early stages of T1D [24]. These findings indicate the importance of IL-1β/IRAK1 signaling in the progressive inflammation of pancreatic islets in children and adolescents with T1D. In addition to studies on PBMCs from T1D patients, multiple lines of evidence have demonstrated the important roles of both CD4+ and CD8+ effector T cells in the immune response to pancreatic islets that drives T1D. Clinically, CD8+ T cells are the most predominant and abundant inflammatory cell type in insulitis. Nevertheless, the activation and propagation of CD8+ T cells and the initiation of β-cell destruction probably require CD4+ T cells [25]. Importantly, the presence of mTOR signaling is necessary for T-helper 1 (Th1) and Th2 effector T cell differentiation [26], and both CD4+ effector T cell types contribute to the pathogenesis of T1D [27]. mTOR is a conserved protein kinase consisting of two distinct complexes: mTOR complex 1 (mTORC1) and mTOR complex 2 (mTORC2). mTORC1 is involved in cell metabolism and cell growth, and the ribosomal protein S6K1 is a well-characterized downstream effector within mTORC1 signaling networks. The production of proinflammatory IL-1β, IL-6, and TNFα cytokines can be induced by stimulating the AKT/mTOR/S6K1 pathway [28], and the phosphorylation of S6K1 by mTORC1 can enhance the efficiency of targeted mRNA translation [29].

The intention of miR-146b is fine-tuning the inflammatory response. Parisi et al. showed that BzATP (ATP analog) is a stimulator of miR-146b in neuroinflammation [30]. On other hand, pathological processes of T1D patients showed low ATP production. Therefore, a vicious circle of low ATP level is formed in the T1D patient. Furthermore, Lu et al. reported that the level of miR-146b was up-regulated after IL-2 treatment. Taken together, IL-2 treatment has been shown to ameliorate T1D pathological processes through miR-146b increase and SRA repression. In this study, we show that IL-2-mediated down-regulation of SRA in T cells is an easy and efficient way to treat T1D pathological development.

The AKT1 network plays physiological roles in the development of human diseases, including developmental and neurological disorders, cardiovascular diseases, cancers, inflammatory autoimmune disorders, and insulin resistance [31]. AKT1 activation requires the phosphorylation of the structural domain of AKT1, located at the position of the catalytic protein kinase core (Thr308), and that of the C-terminal regulatory motif (Ser473) [32]. T308 and S473 are phosphorylated by phosphoinositide-dependent protein kinase 1 (PDK1) and mTORC2, respectively, and the phosphorylation of both residues is required for AKT1 activation and stabilization [33]. In addition, Ras and TRAF6 are involved in the ILR/IRAK1/AKT pathway, and may enhance AKT1 phosphorylation (T308) by promoting AKT1 ubiquitination [34,35]. In this study, we found that lncRNA SRA induced AKT phosphorylation at both the T308 and S473 residues, and that AKT1 phosphorylation negatively regulated PI3K/AKT pathway-mediated Treg function [36]. Based on this evidence, we suggest that increased SRA levels in T1D patients may enhance T effector cell differentiation and diminish Treg-mediated immune regulation.

The metabolic properties between β-cells and CD4+ Tregs facilitate autoimmune T1D. Diabetes has been associated with abnormal lactate metabolism, and a high level of lactate production is a key biological characteristic of T1D. In normal cells, pyruvate is produced through insulin-stimulated glycolysis, and is then converted to lactate. However, long-lasting high lactate levels contribute to a higher insulin resistance status, promoting the development and progression of diabetes [37,38].

Suppression of SRA interrupts long-lasting, lactate-induced metabolic reprogramming, resulting in a multifaceted rescue response that promotes early glycolysis, reduced lactate production, and the progression of oxidative glucose metabolism by coordinating increased uptake and assimilation of glucose. The current study identified the predominant SRA-mediated cell signaling mechanisms likely underpinning this process. Functional shRNA silencing substantially disrupts the effects of SRA, and future work will be aimed at identifying productive metabolic/immune inducer combinations for therapy. Notably, immune inducers, including IL-2, are under clinical investigation, and a recent report has described a promising new IL-2 antibody, F5111.2, with single-agent activity in various preclinical non-obese diabetic (NOD) mouse models of T1D [21]. The use of IL-2 to stimulate Treg-mediated immune regulation is a desirable alternative immunotherapy and early prevention strategy for T1D, according to the findings of a preclinical trial [39] and a phase I/II clinical trial [40]. In our studies, experimental SRA inhibition using shRNA silencing suppressed T1D progression in various metabolic-responsive and β-cell models of T1D. These results add to the accumulating evidence suggesting that the inhibition of SRA may confer therapeutic benefits in T1D and metabolic diseases. Moreover, we discovered that IL-2 therapy directly inhibits SRA, and our findings partially reconcile observations that IL-2 can inhibit T1D progression through anti-SRA-mediated, IRAK1/LDHA/lactate-dependent mechanisms (Figure 7).

SRAs have shown evidence that they directly interact with other proteins and affect protein-related pathways [14]. In addition, through these molecular interactions and the IL-2 mediated pathway, SRA should impact pathological processes of T1D and rescue by the IL-2 mediated pathway. For example, SRA has been reported to bind estrogen receptor (ER) via direct interaction, results in enhanced transactivation of ER activity [14]. Meanwhile, McMurray et al. showed the sex steroid hormone 17β-estradiol (17β-estradiol belongs, together with estrone (E1) and estriol (E3), to the group of sex steroids called estrogens) suppresses the IL-2/IL-2R pathway in CD4+ T cell lines at the transcriptional level [41]. Otherwise, IL-2 regulating a signaling pathway/transcriptional factor might involve in SRA regulation. IL-2 might directly repress SRA via binding ER or others, which offers more information and questions on the potential role of lncRNA SRA and its transcriptional regulation in the clinical course of T1D.

In summary, SRA regulates the IRAK1/LDHA/lactate axis in T1D, and SRA inhibition rescues the oxidative glucose metabolism and cell function of Tregs, as well as apoptosis of β-cells. Our current data suggest a critical, unmet clinical need for treating the most poorly regulated T1D. Future efforts will be focused on elucidating a direct effect of lncRNA SRA-related CD4+ Treg cells on the apoptosis of human β cells (EndoC-βH1) using a co-culture assay, and designing rational combinatorial therapies to maximize the therapeutic effect of SRA/lactate suppression and identify T1D patients who are most likely to benefit from these approaches.

## 4. Materials and Methods

### 4.1. Clinical Participants

Medical records, personal questionnaires, and whole blood samples from each participant were collected after obtaining individual written informed consent. Blood samples from 25 patients diagnosed with T1D at the Children’s Hospital Medical Center in the China Medical University Hospital (CMUH), Taiwan and 26 age-matched, non-hospitalized, healthy volunteers who had no evidence of concurrent infection in the past 6 months were collected at 4 °C and stored at −80°C before use. The study was approved by the Research Ethics Committee of China Medical University and Hospital (CMUH103-REC2-046).

### 4.2. Cell Culture

The MIN6 pancreatic beta cell line [42] and CD4+ MOLT4 T cell lines (used to investigate T cell reactivation) [43] were purchased from the American Type Culture Collection. CD4+ MOLT4 Cells were cultured at 37 °C with 5% CO_2_ in RPMI 1640 medium (HyClone, Thermo Scientific, Taichung, Taiwan), supplemented with 10% fetal bovine serum (FBS; HyClone, Thermo Scientific), 1 mM sodium pyruvate, 100 IU/mL penicillin, and 100 µg/mL streptomycin. MIN6 cells were cultured at 37 °C, with 5% CO_2_ in Dulbecco’s modified Eagle’s medium with 25 mmol/L glucose (HyClone, Thermo Scientific), supplemented with 15% heat-inactivated fetal bovine serum (FBS; HyClone, Thermo Scientific), 100 IU/mL penicillin, and 100 µg/mL streptomycin. The MIN6 cells and CD4+ MOLT4 T cell lines presented in this study were at passages 25–40. To determine the effect of glucose on expression level of studied lncRNAs, cells were exposed to 10 mM (normal conditions, N) and 25 mM glucose (high glucose conditions, H). All assays used MIN6 cells grown to 70–80% confluence unless otherwise stated. Cells were tested regularly for mycoplasma.

### 4.3. LncRNA Profiling

To identify clinically relevant lncRNAs in CD4^+^ Tregs of patients with T1D, we used the Human Disease-Related LncRNA Profiler (cat# RA920D, System Biosciences (SBI, Taichung, Taiwan), consisting of 83 lncRNAs that were selected from the lncRNA database (www.lncRNAdb.org (accessed on 15 January 2021)). The total RNA was isolated from MOLT4 CD4+ Tregs (shCon or shSRA) under high-glucose or normal glucose conditions. Reverse transcription was carried out by using the Moloney murine leukemia virus reverse transcriptase (M-MLV RT) and random primer mix (Thermo Fisher). The values for the cells in high-glucose conditions after normalization by the internal controls served as a basal level of expression of SRA. Delta-delta Ct values (H- versus N-glucose conditions) were used to determine the relative expression as fold changes.

### 4.4. LncRNA and miRNA Expression Analysis

Total RNA was isolated from plasma with TRIzol reagent (Invitrogen, Thermo Fisher Scientific, Taichung, Taiwan). Complementary DNA (cDNA) was generated from total RNA using a High-Capacity cDNA Reverse Transcription kit (ABI, Thermo Fisher Scientific). The expression levels of lncRNAs (SRA, MEG3, and IGF2AS) and miRNAs (miR-146b, miR-148a, miR-203, and miR-103a) were determined using TaqMan noncoding RNA assays and TaqMan miRNA assays (Thermo Fisher Scientific) in a real-time PCR instrument (ABI StepOnePlus, Thermo Fisher Scientific), in accordance with the manufacturer’s protocols.

### 4.5. Vector Construction for Reporter Assays

Human SRA and transcript variant 4 (NR_045587.1; 1473 bp) were cloned into the pmirGLO luciferase vector (Promega, Madison, WI, USA) at restriction sites PmeI and XhoI. The entire lncRNA SRA sequence with its two predicted binding sites, seed locations 219 and 1122 (for miR-146b), was cloned into the pmirGLO luciferase vector (Promega) as short and long wild-type (WT) constructs, respectively. The mutation of lncRNA SRA sequences at the predicted binding sites was performed with sense/antisense pairs of mutagenic oligonucleotide primers and a QuikChange Lightning Site-Directed Mutagenesis kit (Agilent, Clara ,CA, USA). The DNA sequences of all constructs were verified by Sanger sequencing.

### 4.6. miRNA Mimic Reporter Assay and Antagomir Transfection

Single-stranded, chemically enhanced oligonucleotides, scrambled mimics, and miR-146b mimics designed for the overexpression of mimic miRNA and knockdown of miRNA were purchased (miRIDIAN miRNA mimics, Dharmacon, Taichung, Taiwan). Cells were seeded in six-well plates at 50% confluence and allowed to attach for 24 h. Co-transfection was performed with the scrambled mimic (100 nM), miR-146b mimic (100 nM), miR-146b antagomir (10, 50 and 100 nM), and the constructed lncRNA SRA reporter vector using Lipofectamine 2000 (Invitrogen). Cell extracts were harvested at 48 h post-transfection, and the luciferase activity was measured using the Dual-Luciferase Reporter Assay System (Promega, Taichung, Taiwan). The procedures were carried out in accordance with the approved guidelines.

### 4.7. shRNA Knockdown

CD4+ MOLT4 T cells were infected with lentiviral particles expressing shRNA targeting SRA in the presence of 8 μg/mL protamine sulfate for 24 h, and then selected for 48 h using puromycin (2 μg/mL). shLacZ (TRCN0000231726), which targets the lacZ gene, was used as a control, shSRA (human: TRCN0000255575 and TRCN0000265668; mouse: TRCN0000351198 and TRCN0000334446; National RNAi Core Facility, Academia Sinica, Taiwan). The knockdown efficiency of lncRNA SRA and miR-146b was assessed using TaqMan assays, as described previously.

### 4.8. Immunoblotting

Cellular proteins were extracted using RIPA lysis buffer (Thermo Fisher Scientific) supplemented with 2 mM NaVO_3_, 1 mM PMSF, and 5 mM NaF. The plasma and cellular total protein concentrations were quantified by a Pierce BCA Protein Assay kit (Thermo Fisher Scientific). Primary antibodies against IRAK1, AKT, phospho-AKT (Ser473), S6 kinase, phospho-S6 kinase (Ser235/236), and β-actin (1:1000) (Cell Signaling Technology, CST) were used in the analysis. The protein expression level was assessed using an Immobilon Western Chemiluminescent HRP Substrate kit (Millipore, Merck, Taichung, Taiwan) in an ImageQuant LAS4000 Biomolecular Imager (GE Healthcare, Taichung, Taiwan).

### 4.9. Immunofluorescence

Cells were fixed with 4% paraformaldehyde and incubated overnight with IRAK1 antibody (CST, #4504S, 1/100), LDHA (CST, #3582S, 1/100) or phosLDHA-Tyr10 (CST, #8176S, 1/100). Alexa Fluor 488 anti-rabbit antibody (Invitrogen Life Technologies, 1/250, (Taichung, Taiwan) was used as the secondary antibody. Nuclei were stained with Hoechst 33342 (CST, #4082, 1/1000). Five images of each sample were collected with a fluorescence microscope (BioTek, Lionheart FX, Taipei, Taiwan). Two channels were acquired sequentially with the following excitation and emission parameters: (488 nm, 540 nm) for the IRAK1/LDHA/phosLDHA-Tyr10 signal, and (355 nm, 465 nm) for Hoechst 33342. Data processing and quantification of the fluorescence signal were performed with Gen5.

### 4.10. Analysis of Apoptosis, ROS, and ATP

Cells were seeded in a 24-well plate in RPMI. The next day, the medium was refreshed. After 48 h of incubation, apoptosis was evaluated by flow cytometry using the Fluorescein isothiocyanate (FITC) annexin V apoptosis detection kit and propidium iodide (PI) (Biolegend, Taichung, Taiwan). Fluorescence activated cell sorter (FACS) analysis was performed on 10,000 events with a BD Accuri C6 Plus system (Taichung, Taiwan) [44]. ROS content was assessed using a MAK-143 intracellular ROS kit, following the manufacturer’s protocol (Sigma-Aldrich, Taichung, Taiwan). Cells were cultured in 96-well black plates at a density of 5 × 10^4^ cells per well overnight. Briefly, 100 µL/well of Master Reaction Mix was added to the cell plate and incubated for 1 h in an incubator. ATP was measured from cells using a BioTracker ATP-Red Live cell dye (Sigma-Aldrich), according to the manufacturer’s protocol. Data were collected using a flow cytometer (BD Accuri C6 Plus).

### 4.11. Analysis of Lactate Secretion

Lactate content in growth medium was assessed using an enzyme-based bioluminescent kit (Promega, Taichung, Taiwan) and colorimetric/fluorometric assay kit (Biovision, #607, Taichung, Taiwan), according to the manufacturer’s protocol. The lactate of the cell culture supernatant was measured by mass spectrometry (MS), and cell culture supernatant (40 µL) or plasma samples (40 µL) were mixed with 5% trichloroacetic acid (160 µL). The mixture was vortexed for 30 seconds and kept on ice for 5 min (vortex twice/1 minute). Samples were centrifuged at 13,000× *g* for 30 min (4 °C), and the supernatants were transferred to vials for MS analysis on a 3000 QTRAP mass spectrometer equipped with an ESI ion source operated in negative ion mode. Instrument control, data acquisition, and processing were performed using Analyst 1.5.2 software. First, collision-induced dissociation (CID) experiments of lactic acid standards were performed in product ion scan mode. Then, MS data were acquired in the multiple reaction monitoring (MRM) mode.

### 4.12. IL-2 Treatment

The recombinant human IL-2 protein (rhIL-2) was purchased from PeproTech (catalog# 200-02-50, lot# 101712-1, Taichung, Taiwan). A final rhIL-2 concentration of 200 units/mL for 48 h of treatment was used in the analysis.

### 4.13. Statistical Analysis

The mean and standard deviation were calculated for each of the investigated parameters. Error bars represent the standard deviation (SD) of a triplicate set of experiments. Statistical analyses were performed using an unpaired Student’s *t*-test. The level of statistical significance was set at *p* < 0.05.

## Figures and Tables

**Figure 1 ijms-22-01720-f001:**
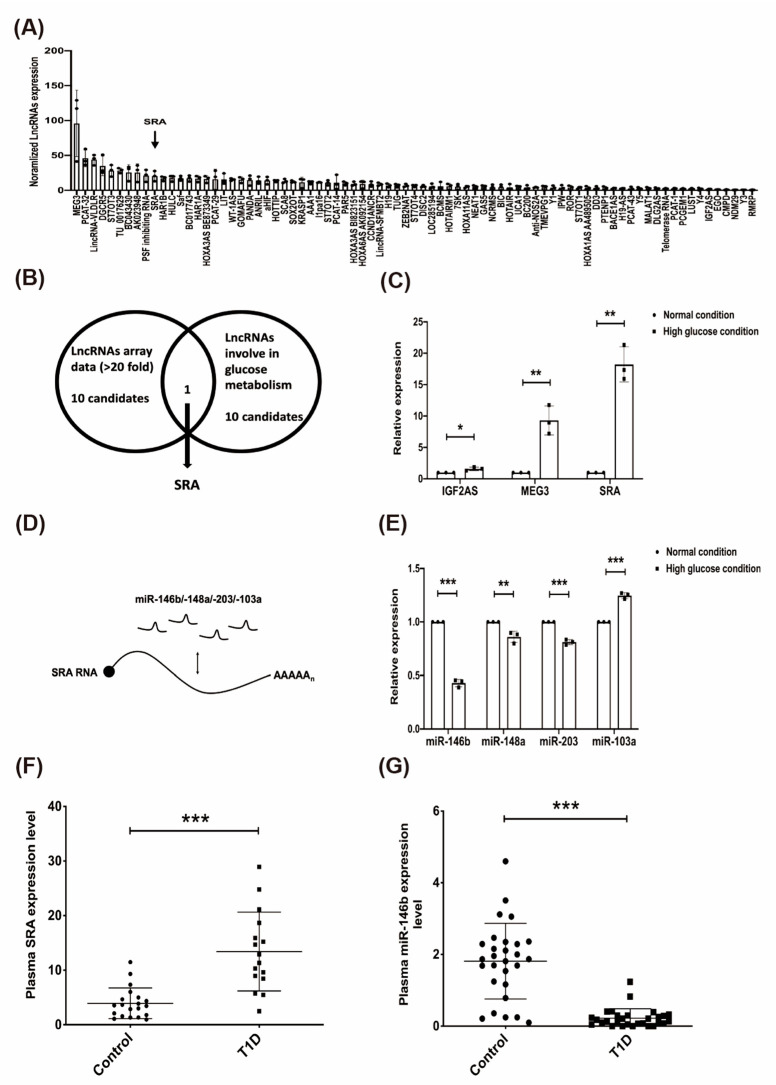
The effect of high glucose on candidate long non-coding RNAs (lncRNAs)/miRNAs in CD4+ MOLT4 Tregs and expression of lncRNA steroid receptor RNA activator (SRA)/miR-146b in the plasma of patients with type 1 diabetes mellitus (T1D). (**A**) Deregulated lncRNAs in high glucose (H) concentrations for 72 h (N: normalized to normal conditions). Each bar represents the expression of lncRNAs relative to that of the internal control. Cells were exposed to N/H conditions for 72 h. After treatment, RNA was isolated, and a disease-related lncRNA array was performed. (**B**) Venn diagram shows the high-glucose-induced lncRNAs (>two-fold, 15 lncRNAs, Appendix A) in CD4+ MOLT4 Tregs and selected glycolysis-related lncRNAs from the literature (10 lncRNAs, Appendix A). In total, (**C**) three lncRNAs (IGF2AS, MEG3, and SRA) and (**E**) four miRNAs (miR-146b, miR-148a, miR-203, and miR-103a) were assessed using real-time qPCR. * significant changes compared with the control (*p* < 0.001). (**D**) Schematic representation of the competing endogenous RNA circuitry linking lncRNA-SRA and miR-146b/-148a/-203/-103a (Appendix A). The plasma expression levels of (**F**) lncRNA SRA and (**G**) miR-146b in the control subjects and the patients with T1D. The plasma levels of lncRNA SRA and miR-146b were determined using real-time qPCR. Mean ± SD, Student’s *t*-test (unpaired, two-sided). * *p* < 0.05, ** *p* < 0.01, *** *p* < 0.001.

**Figure 2 ijms-22-01720-f002:**
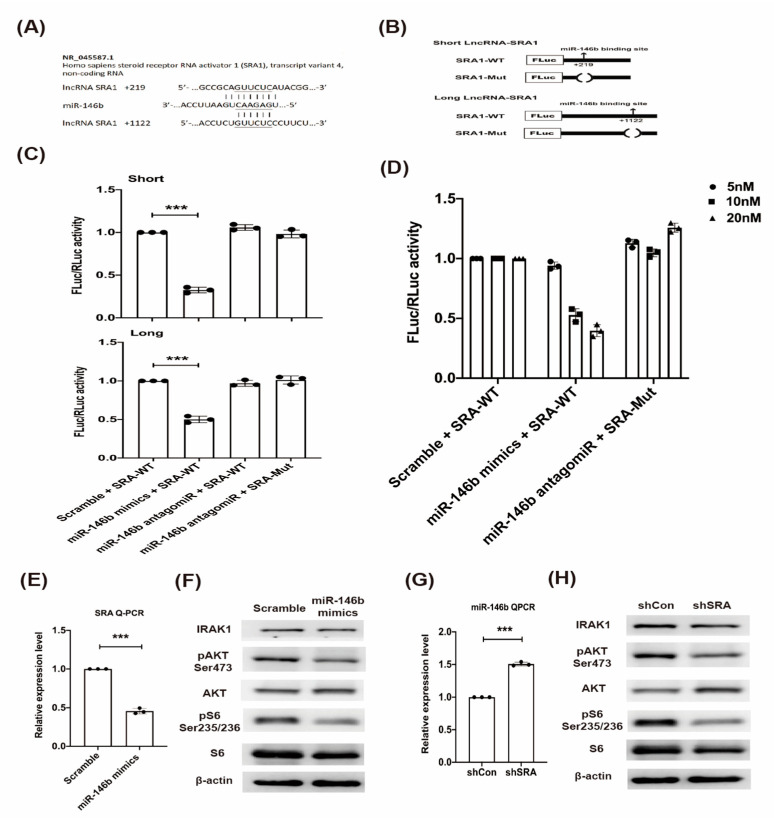
LncRNA SRA is direct target of miR-146b and regulates interleukin-1 receptor-associated kinase 1 (IRAK1)/protein kinase B (AKT)/ribosomal protein S6 kinase 1 (S6K1) signaling. (**A**) Two putative miR-146b miRNA response elements (MREs) (seed locations 219 and 1122) were predicted in the 3′-untranslated regions (UTRs) of lncRNA SRA. (**B**) The short and long lncRNA SRA sequences (SRA-WT (wild type)), in parallel with mutant derivatives lacking MREs (SRA-Mut), were cloned downstream of the firefly luciferase coding region. (**C**) The luciferase activity of the short and long SRA-WT and SRA-Mut constructs co-transfected with the scrambled-miR control, miR-146b mimic, and miR-146b antagomir was assessed using a luciferase reporter assay at 48 h post-transfection. (**D**) The dose-dependent effects of the scrambled-miR control, miR-146b mimic, and miR-146b antagomir on the SRA-WT construct were assessed. (**E**) The inhibitory effect of a transfected miR-146b mimic on endogenous SRA expression. (**G**) The induction of endogenous miR-146b expression by the knockdown of SRA in T cells. The effect of (**F**) miR-146b mimic transfection and (**H**) SRA knockdown on IRAK1/AKT/S6K1 signaling in CD4+ T cells. Mean ± SD, Student’s *t*-test (unpaired, two-sided). *** *p* < 0.001.

**Figure 3 ijms-22-01720-f003:**
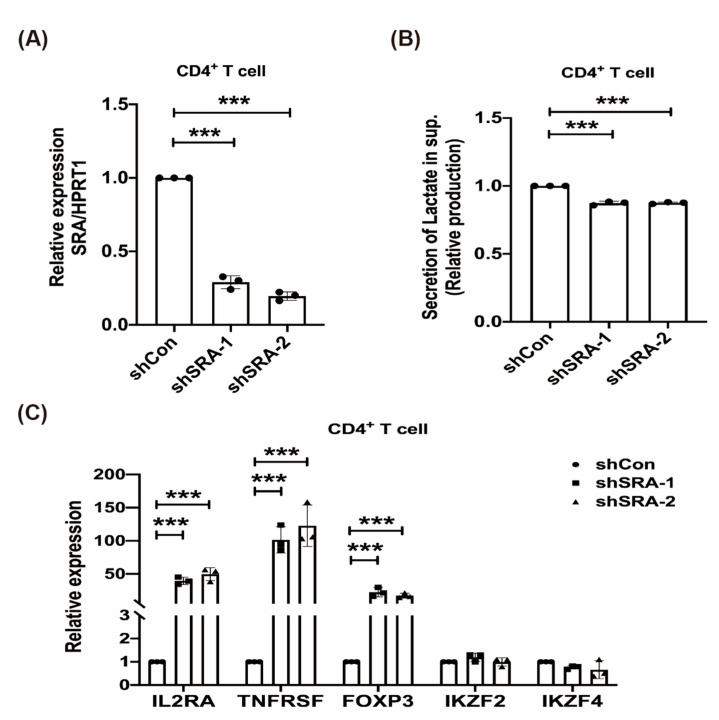
LncRNA SRA suppresses CD4+ Treg function. (**A**) LncRNA SRA expression analysis of CD4+ MOLT4 Tregs (shCon and shSRA). (**B**) SRA induces lactate metabolism. Lactate was quantified in CD4+ MOLT4 Tregs (shCon and shSRA) and (**C**) mRNA expression analysis of Treg signature genes (shCon and shSRA). SRA silencing caused a sharp increase in the amount of mRNA of genes that promoted Treg function. Mean ± SD, Student’s *t*-test (unpaired, two-sided). *** *p* < 0.001.

**Figure 4 ijms-22-01720-f004:**
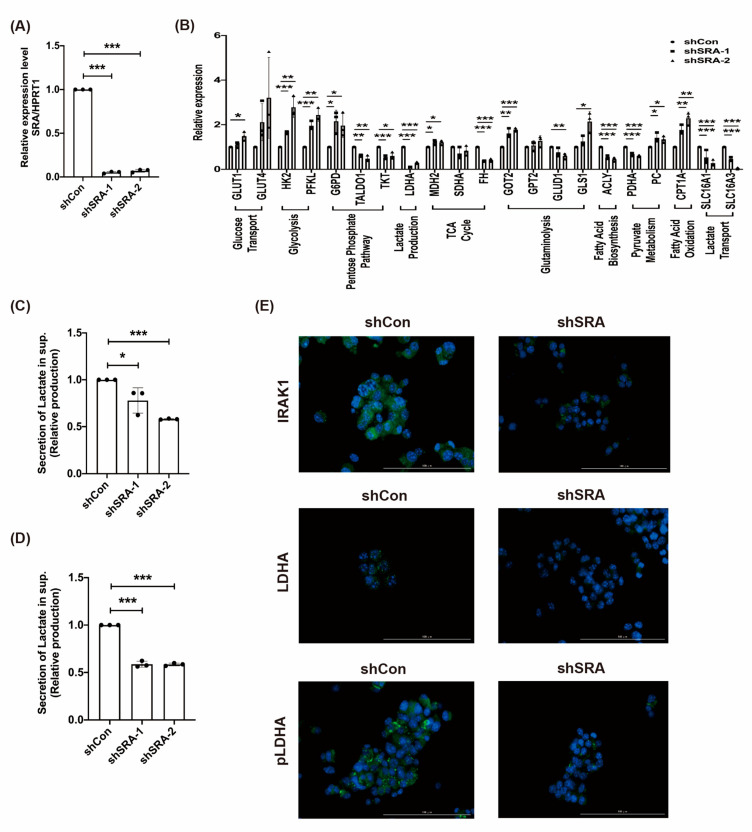
lncRNA SRA induces MIN6 β-cell metabolic reprogramming and the IRAK1/lactate dehydrogenase A (LDHA)/phosphorylated LDHA (pLDHA) signaling pathway. (**A**) lncRNA SRA expression analysis of MIN6 β-cells (shCon and shSRA). (**B**) mRNA expression analysis of metabolic genes after knockdown with lncRNA SRA (shCon and shSRA). Lactate was analyzed by mass spectrometry (MS) (**C**) or a luminescence-based kit (**D**) in MIN6 β-cells after knockdown lncRNA SRA (shCon and shSRA). Representative immunofluorescence images of shCon and shSRA MIN6 β-cells. Co-staining for (**E**) IRAK1, LDHA, or phosphorylation of LDHA (Tyr10) (green) with DAPI (blue). Scale bar, 100 µm. Mean ± SD, Student’s *t*-test (unpaired, two-sided). * *p* < 0.05, ** *p* < 0.01, *** *p* < 0.001.

**Figure 5 ijms-22-01720-f005:**
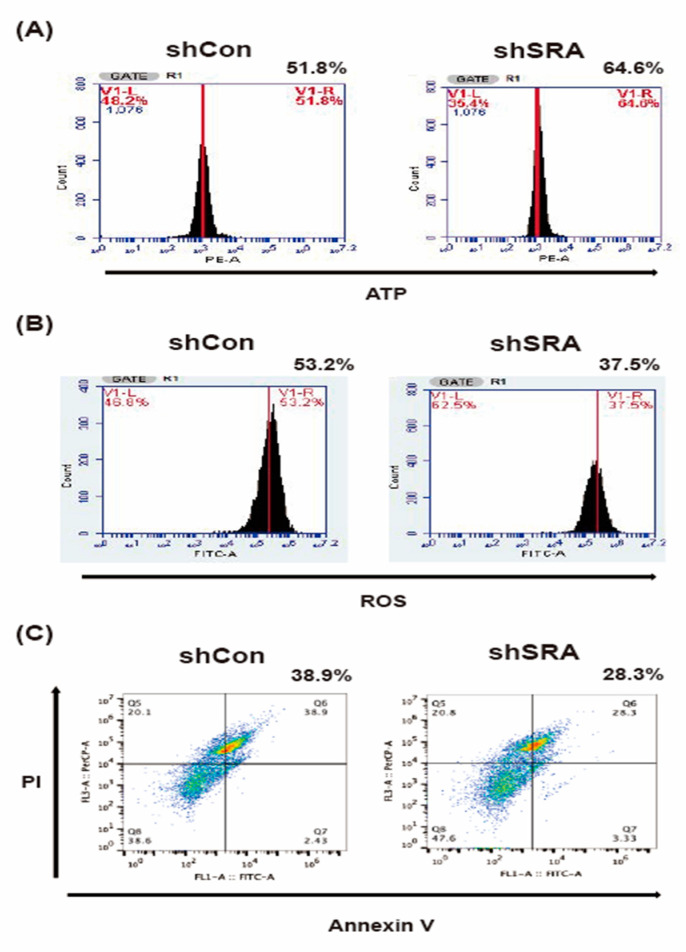
lncRNA SRA silencing induces ATP production, represses reactive oxygen species (ROS) levels, and inhibits MIN6 β-cell apoptosis. Representative flow cytometric analysis for (**A**) ATP production, (**B**) ROS induction, (**C**) and apoptotic markers by annexin V and propidium iodide (PI) staining in MIN6 β-cells after knockdown lncRNA SRA (shCon and shSRA). Mean ± SD, Student’s *t*-test (unpaired, two-sided).

**Figure 6 ijms-22-01720-f006:**
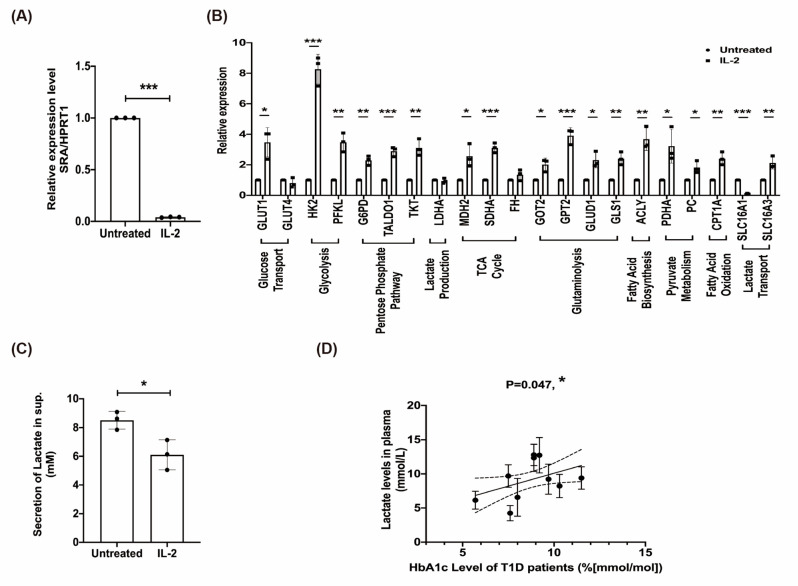
The effect of recombinant human IL-2 (rhIL-2) treatment on the expression of lncRNA SRA, metabolic pathways in MIN6 β-cells, and lactate associated with poor clinical outcome. The cellular expression of (**A**) SRA was assessed in cells using real-time qPCR after 48 h of treatment with rhIL-2 (200 units/mL) and compared with that of the untreated control cells. * significant changes compared with the control or untreated cells (*p* < 0.001). (**B**) Real-time qPCR analysis showed the metabolic pathway in shSRA versus shCon-transduced MIN6 β-cells. (**C**) Lactate production of rhIL-2 treatment (300 units/mL) for MIN6 β-cells. (**D**) Correlation between lactate level and HbA1c level in plasma from patients with T1D. Mean ± SD, Student’s *t*-test (unpaired, two-sided). * *p* < 0.05, ** *p* < 0.01, *** *p* < 0.001.

**Figure 7 ijms-22-01720-f007:**
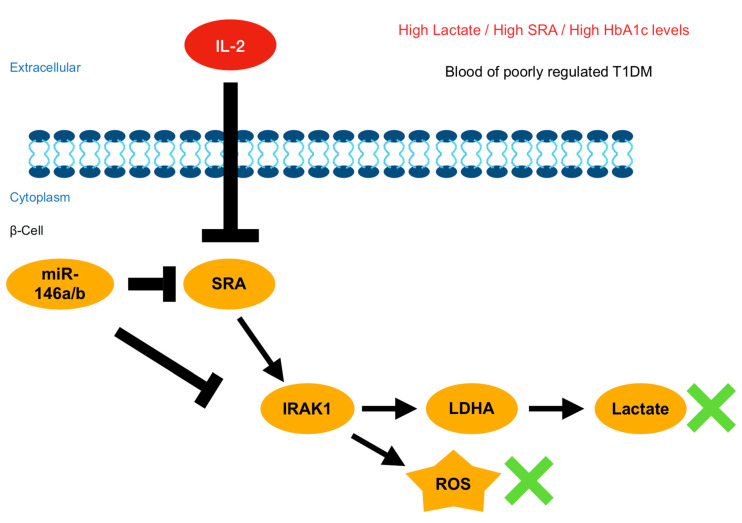
Schematic diagram summarizes aberrant lncRNA SRA-mediated networks in β-cells of T1D. The panel indicates that the aberrant expression of IRAK1 and LDHA by lncRNA SRA induction through competing miR-146b and driving metabolic reprogramming and IL-2 treatment represses the lncRNA SRA-mediated IRAK1/LDHA signaling pathway. (green x means products inhibition)

**Table 1 ijms-22-01720-t001:** Clinical characterization of T1DM patients (*n* = 25).

Parameter	First Onset	Recent Onset (Enrolled)
Gender, male/female = 10/15		
Age, mean (SD)	9.5 (4.8)	18.4 (5.2)
Fasting glucose level, mg/dL (SD)	345.2 (132.6)	186.0 (89.3)
Hemoglobin A1c (HbA1c) level, % (SD)	12.7 (2.5)	9.0 (1.8)
C-peptide, ng/mL (SD)	0.3 (0.2)	0.4 (0.2)
Hemoglobin, g/dL (SD)	14.2 (1.7)	14.2 (1.6)
Hematocrit, % (SD)	40.4 (3.7)	40.6 (4.4)
Red blood cells, 10^6^ cells/L (SD)	5.3 (0.6)	5.4 (0.8)
White blood cells, 10^3^ cells/mm^3^ (SD)	14.4 (8.2)	14.1 (7.6)
Platelets, 10^3^ cells/mm^3^ (SD)	328.1 (106.2)	327.1 (99.8)
Neutrophils, % (SD)	68.9 (19.5)	64.5 (21.2)
Lymphocytes, % (SD)	25.9 (17.6)	28.9 (20.7)
Monocytes, % (SD)	5.6 (2.7)	6.4 (3.0)
Total cholesterol, mg/dL (SD)	198.1 (45.2)	190.1 (37.6)
High-density lipoprotein, mg/dL (SD)	60.2 (18.3)	62.3 (19.0)
Low-density lipoprotein, mg/dL (SD)	119.2 (37.6)	104.2 (29.1)
Triglyceride, mg/dL (SD)	73. (68.5)	105.8 (133.3)

## Data Availability

The data presented in this study are available in the Appendix A.

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
