# Peer review of "Long, Noncoding RNA SRA Induces Apoptosis of β-Cells by Promoting the IRAK1/LDHA/Lactate Pathway"

_ijms, 2021, doi:10.3390/ijms22041720_

Round 1
Reviewer 1 Report
In the submitted paper by Y Huang et al., the authors investigated the long non-coding RNA steroid receptor (LncRNA SRA) regulation in b-cell of Type 1 diabetes mellitus, characterized by hyperglycemia-induced cellular damage and decreases in regulatory T cells.
The data demonstrates that the LncRNA SRA-mediated IRAK1/LDHA/pLDHA/lactate pathway in b-cells is involved in the progression of Type 1 diabetes mellitus. Compared to normal conditions, high glucose conditions increased the level of LncRNA SRA in the cells. Additionally, the expression level of LncRNA SRA was significantly higher in the Type 1 diabetes mellitus patients than in the heathy control. Moreover, experimental SRRA inhibition using shRNA silencing rescues the oxidative glucose metabolism and cell function of T cells and apoptosis of b-cells.
These are nice findings and the data are well presented. While the subject of this research article is highly worthy of investigation, the content of this submission is purely descriptive. In order to explore the role of the LncRNA SRA-mediated IRAK1/LDHA/pLDHA/lactate pathway, I invite the authors to use an animal model of Type 1 diabetes mellitus.
Author Response
Response to Reviewer 1 Comments
Point 1: These are nice findings and the data are well presented. While the subject of this research article is highly worthy of investigation, the content of this submission is purely descriptive. In order to explore the role of the LncRNA SRA-mediated IRAK1/LDHA/pLDHA/lactate pathway, I invite the authors to use an animal model of Type 1 diabetes mellitus.
Response 1: Thank you for the critical and important comments. Several inbred mice models of T1D are used for investigating pathogenic complexity of autoimmune diabetes, such as spontaneous models, transfer models, humanized mouse models, pathogen (lymphocytic choriomeningitis virus glycoprotein)-induced models, and alloxan and streptozotocin (STZ) pharmacological models. Selection of the best mice model in this study is an important in vivo design for elucidating the molecular mechanism of lncRNA SRA-induced apoptosis in β cells. However, each experimental model has differing strengths and weaknesses to unravel the special topic involving in the immune signaling pathways of T1D pathologic development and treatment. We appreciate that you point out the weakness of the current findings and give us an important suggestion to present a consummate elucidation in this study. The planning of animal models would be the next work following this in vivo and clinical study, which offer more information and question on the potential role of lnRNA SRA and its signalling in the clinical course of T1D.
Reviewer 2 Report
The study by Huang et al. explores the role SRA in type 1 diabetes. It explores using human samples and cell cultures the effect of SRA inhibition on the progression of type 1 diabetes. This study utilizes novel technologies to nuance the relationship of lcRNAs and miRNA interactions in cells to understand how this interaction can be targeted and utilized for pharmacological benefit.
General Items:
- Please add the data points in the graphs. Newer publications are requiring this format and it gives the readers a sense of the true data distribution. In addition could you include the sample size for testing, it looks like you only had an n=3 from your statistical experimentation, which may not be enough based on the distribution of the SD in some of the experiments.
Specific Items:
- IL2 - while an important mediator in the chain of events, it is unclear if IL2 directly mediates a reduction in SRA availability. There have been no protein complex assays that suggest so, and therefore, leaves me to wonder if IL2 functions in aberrant from what is described to result in the diminished SRA.
- miR146 - Could you comment on what are normal stimulators of miR146b. Why would it be down-regulated in T1D? Is it known if IL2 could also stimulates its production?
- Figure 7 does not seem to follow the story of the study. The reading left me under the impression that miR146 inhibits SRA and thus inhibits the chain of events, but the figure indicates otherwise.
- Please address the lack of co-culture conditions (i.e. pancreatic cells & t-cells)?
Author Response
Response to Reviewer 2 Comments
Point 1: Please add the data points in the graphs. Newer publications are requiring this format and it gives the readers a sense of the true data distribution. In addition could you include the sample size for testing, it looks like you only had an n=3 from your statistical experimentation, which may not be enough based on the distribution of the SD in some of the experiments.
Response 1: Thank you for the valuable comments. As suggested by the reviewer that add the data points in the graphs. We have reproduced and add data in the figures (Fig 1A, 1C, 1E, 2C, 2D, 2E, 2G, 3A, 3B, 3C, 4A, 4B, 4C, 4D, 6A, 6B and 6C) for represents data distribution in the revised manuscript.
Point 2: IL2 - while an important mediator in the chain of events, it is unclear if IL2 directly mediates a reduction in SRA availability. There have been no protein complex assays that suggest so, and therefore, leaves me to wonder if IL2 functions in aberrant from what is described to result in the diminished SRA.
Response 2: Thank you for the valuable comments. It is good suggestion for future work about IL-2 how directly mediates SRA. Based on references, SRA have showed evidences that directly interacts with other protein and affects the protein-related pathways. In addition, through these molecular interactions and IL-2 mediated pathway from references, SRA should impact pathological processes of T1D and rescue by IL-2 mediated pathway. For example, SRA has been reported to binds Estrogen receptor (ER) via direct interaction, results in enhances transactivation of ER activity1. Meanwhile, McMurray et al. showed the sex steroid hormone 17β-estradiol (17β-estradiol belongs together with Estrone (E1) and Estriol (E3) to the group of sex steroids called Estrogens) suppresses IL-2/IL-2R pathway in CD4+ T cell lines at the transcriptional level2. Otherwise, IL-2 regulated signaling pathway/Transcriptional factor might involving in SRA regulation. The valuable comments point out next work that IL-2 might directly represses SRA via binding ER or others. The planning of molecular mechanism would be the next work following this clinical study, which offer more information and question on the potential role of lnRNA SRA and its transcriptional regulation in the clinical course of T1D.
References:
1.Lanz RB, McKenna NJ, Onate SA, Albrecht U, Wong J, Tsai SY, et al. A steroid receptor coactivator, SRA, functions as an RNA and is present in an SRC-1 complex. Cell (1999) 97:17–27.
2.McMurray RW, Ndebele K, Hardy KJ, Jenkins JK. 17-β-estradiol suppresses IL- 2 and IL-2 receptor. Cytokine. 2001;14(6):324–33.
Point 3: miR146 - Could you comment on what are normal stimulators of miR146b. Why would it be down-regulated in T1D? Is it known if IL2 could also stimulates its production?
Response 3: Thank you for the valuable comments. The intention of miR-146b is fine-tuning the inflammatory response. Parisi et al. showed BzATP (ATP analog) is a stimulator of miR-146b in neuroinflammation1. On other hand, pathological processes of T1D patients showed low ATP production in our data (Fig 5A). Therefore, a vicious circle of low ATP level is formed in the T1D patient. Furthermore, Lu et al. reported the level of miR-146b was up-regulated after IL-2 treatment. Taken together, the IL-2 treatment has been shown to ameliorate T1D pathological processes through miR-146b increase and SRA repression. In this study, we show that IL-2-mediated down-regulation of SRA in T cells is an easy and efficient way for treatment of T1D pathological development.
References:
1.Parisi C., Arisi I., D’Ambrosi N., Storti A.E., Brandi R., D’Onofrio M., Volonté C. Dysregulated microRNAs in amyotrophic lateral sclerosis microglia modulate genes linked to neuroinflammation. Cell Death Dis. 2013;4:e959.
2.Lu Y, Hippen KL, Lemire AL, Gu J, Wang W, Ni X, et al. miR-146b antagomir-treated human Tregs acquire increased GVHD inhibitory potency. Blood (2016) 128(10):1424–35.
Point 4: Figure 7 does not seem to follow the story of the study. The reading left me under the impression that miR146 inhibits SRA and thus inhibits the chain of events, but the figure indicates otherwise.
Response 4: Thank you for the valuable comments. As suggested by the reviewer that we have reproduced the Fig 7, swap the positions of SRA and miR-146a/b to fit the story line. We appreciate that you point out the weakness of the Fig 7 and give us an important suggestion to present a consummate elucidation in this study.
Point 5: Please address the lack of co-culture conditions (i.e. pancreatic cells & t-cells)?
Response 5: Thank you for the valuable comments. As suggested by the reviewer the details on co-culture conditions has been included in methods section under methods section 4-13, “Pancreatic beta cells and CD4+ T cells co-cultures” -Pancreatic beta cells were plated at 2x105 cells per well in a 6-well culture plates in DMEM medium and allowed to adhere overnight at 37°C in a humidified incubator and 5% CO2 atmosphere. After 24 hours, 2x106 CD4+ T cells were added to pancreatic beta cells in transwell system. Total RNA of T cells were harvested and evaluated by qPCR analysis.
Round 2
Reviewer 1 Report
Ok.
Author Response
Response to Reviewer 1 Comments
Point 1: OK
Response 1: Thank you for the valuable comments. We appreciate your kindly comments and suggestions on our manuscript.

Reviewer 2 Report
It was not clear from my second review how and when the co-culture studies were implemented in your study or just listed as experiments.
Secondly based on the two figures, I am unsure, which model you believe your work has addressed.
Author Response
Response to Reviewer 2 Comments
Point 1: It was not clear from my second review how and when the co-culture studies were implemented in your study or just listed as experiments.
Secondly based on the two figures, I am unsure, which model you believe your work has addressed.
Response 1: Thank you for the valuable comments. We appreciate your kindly comments and suggestions on our manuscript. Here we apologize that we misunderstood your question “Point 5: Please address the lack of co-culture conditions (i.e. pancreatic cells & t-cells)? “and added unnecessary methodology “ 4.13. Pancreatic beta cells and CD4+ T cells co-cultures” in the 1st revision. Therefore, we need to clarify that we did not have any co-culture experimental design in the study, and we have removed this information relative to co-culture assay in the 2nd revised manuscript. However, we have added your important suggestion in our future work (Also refer line 397-399 in the last paragraph of the Conclusion).
We admit that we cannot make sure your comment on which figures that we need to describe more carefully and detailly. But we attempted to incorporate the responses to your question 2 and 3 in the revised manuscript (also refer red line 334-342 and 382-393 in the Discussion section). If possible please give us more informative comment and suggestion as you mentioned above. Thank you again.
